# DIFFERENTIABLE OPTIMIZATION OF SIMILARITY SCORES BETWEEN MODELS AND BRAINS

**Nathan Cloos**[1]**, Markus Siegel**[2]**, Scott L. Brincat**[1]**, Earl K. Miller**[1]**, Christopher J. Cueva**[1]
[1]Department of Brain and Cognitive Sciences, MIT
[2]Dept. of Neural Dynamics and Magnetoencephalography, HIH Tübingen
`{nacloos,sbrincat,ekmiller,cjcueva}@mit.edu`
`markus.siegel@uni-tuebingen.de`

## ABSTRACT

What metrics should guide the development of more realistic models of the brain? One proposal is to quantify the similarity between models and brains using methods such as linear regression, Centered Kernel Alignment (CKA), and Procrustes distance. To better understand the limitations of these similarity measures we analyze neural activity recorded in two experiments on nonhuman primates, and optimize synthetic datasets to become more similar to these neural recordings. How similar can these synthetic datasets be to neural activity while failing to encode task relevant variables? We find that some measures like linear regression and CKA, differ from Procrustes distance, and yield high similarity scores even when task relevant variables cannot be linearly decoded from the synthetic datasets. Synthetic datasets optimized to maximize similarity scores initially learn the first principal component of the target dataset, but Procrustes distance captures higher variance dimensions much earlier than methods like linear regression and CKA. When optimizing data towards neural activity we find the similarity scores at which principal components of this neural activity are learned, is well predicted by replacing neural activity with random datasets with matching distributions of singular values, suggesting new theoretical directions for studying these similarity measures.

## 1 INTRODUCTION

Natural intelligence entails the interactions between many systems: perception, cognition, action, memory, etc. and so ultimately many of the open questions in systems neuroscience will require models that bridge these systems. However, there are many challenges as we work towards these multi-system models of the brain. We need guidance to search through this huge design space of potential models. It would be helpful to have metrics that allow us to extract general principles that are applicable across many models, or alternatively, highlight the power of heterogeneity, and shine a spotlight on combinations of tasks, datasets, and models that are not currently well explained.

In this work we study several popular methods that have been proposed to quantify the similarity between models and neural data, in particular, linear regression (Yamins et al., 2014; Schrimpf et al., 2018), Centered Kernel Alignment (CKA) (Kornblith et al., 2019), and angular Procrustes distance (Williams et al., 2021; Ding et al., 2021). See appendix A.1 for a brief overview. We analyzed neural data from two studies on nonhuman primates (Figure 1) and compared the neural responses to synthetic datasets with different similarity scores. In order to study what drives high similarity scores we directly optimize the synthetic datasets to maximize their similarity to the neural datasets as assessed by different methods, for example, linear regression, CKA, or angular Procrustes distance.

## 2 RESULTS

### 2.1 HIGH SIMILARITY SCORES FAILING TO ENCODE TASK VARIABLES

We start by asking if synthetic datasets with high similarity scores relative to the neural data, encode task relevant variables, for example, the stimulus features or the response of the monkey, in the

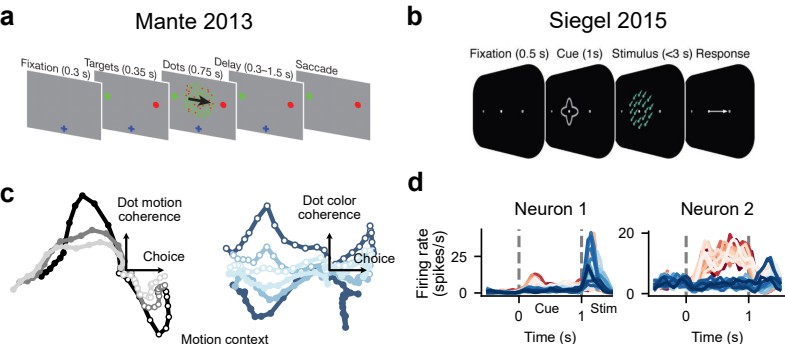

Figure 1: Two neural datasets analyzed in this paper. To study similarity measures we computed similarity scores between artificial datasets and electrode recordings from (a) prefrontal cortex (PFC) from Mante et al. (2013) and (b) ventral stream V4 from Siegel et al. (2015) in monkeys performing an experimental task that required the animal to attend to either color or motion information while ignoring the non-cued feature of the stimuli. On each trial, a field of colored moving dots is shown. Monkeys are given a cue at the beginning of the trial to determine whether the dots in the stimulus are moving left vs right, or are red vs green. The monkey reported its choice with a saccade to one of two visual targets. In both datasets we analyzed neural activity taken when the dot stimulus was presented. (c) Neural activity is visualized in a low-dimensional space capturing task-relevant dynamics using the targeted dimensionality reduction method from Mante et al. 2013. Each curve shows the average neural activity for a different experimental condition. See Mante et al. 2013 for a detailed description of the analysis. This visualization highlights features of the data but the similarity scores were computed using the firing rates from the electrode recordings before any dimensionality reduction. (d) Neural firing rates for two example neurons are shown with the colors denoting average activity for different experimental conditions.

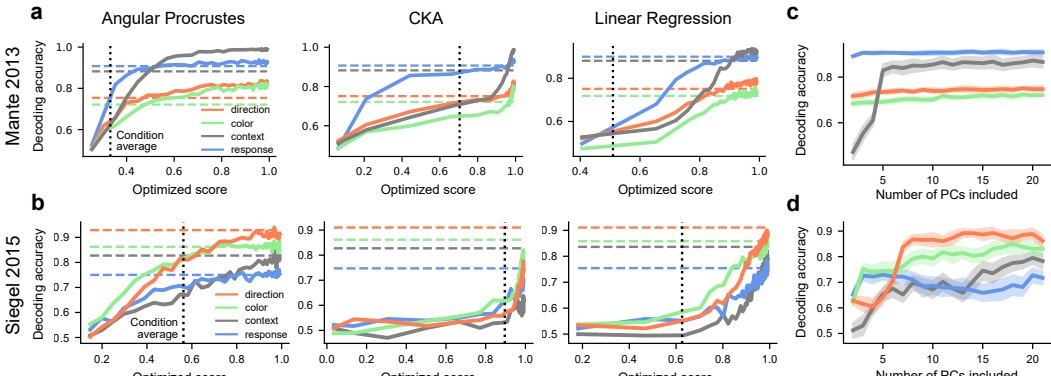

Figure 2: (a, b) Decode accuracy for experimental variables versus similarity scores. Decode is from synthetic data optimized towards greater similarity with the neural data from (a) Mante et al. (2013) and (b) Siegel et al. (2015). Horizontal dashed lines indicate the decode accuracy from the neural data. (c, d) Decode accuracy from neural data versus number of principal components included in the decode. Decode uses data from (c) prefrontal cortex from Mante et al. 2013 and (d) V4 from Siegel et al. 2015.

same way as the neural data. More specifically, is it possible for the synthetic datasets to have a high similarity score while failing to encode task relevant variables? Surprisingly we find that for linear regression and CKA the answer is yes, a high similarity score does not necessarily mean the synthetic datasets encode task relevant variables like the neural data. Figures 2a and 2b show the decode accuracy of a linear classifier trained to decode task relevant variables (cross-validated across different conditions) as the similarity score increases. Before optimization, the synthetic datasets initially consisted of Gaussian noise and the decode accuracy was near the baseline chance level of 0.5 as expected for the binary classifier used in this analysis. Consider the synthetic data optimized towards the Siegel 2015 neural recordings using CKA similarity (second row and column

in Figure 2). When the synthetic dataset has a high similarity score of 0.9 the decode accuracy for all the task variables is still less than that found in the neural activity (horizontal dashed lines). This is in contrast to the case where synthetic data is optimized to maximize angular Procrustes similarity (first column) and a similarity score of 0.9 yields a dataset that encodes task variables to the same degree as the neural recordings. Note that both CKA and angular Procrustes have the same similarity scale ranging between 0 and 1 (perfect similarity). See appendix A.1 for more details. As another example, consider the decode accuracy for the color of the stimulus that was shown in the experiments (green curves). For linear regression and CKA the similarity score can be high even when the data does not encode this task variable at the level found in the neural activity, suggesting there is a mismatch between the datasets with a high similarity score and features of the neural activity. In contrast, for the angular Procrustes distance the synthetic datasets that have a high similarity score relative to the neural activity also encode the task relevant variables.

As a final more optimistic note, for all the similarity measures, we can see a general trend that optimizing the synthetic data towards higher similarity eventually leads the data to encode task relevant variables above the baseline chance level of 0.5 accuracy. One perspective on this decoding analysis is that it gives a window into what is a 'high' similarity score. If a high score is one where the data is encoding task relevant variables then these similarity numbers will need to be interpreted differently based on the similarity measure.

A specific number for the similarity score, 0.9 for example, can have different interpretations depending on the measure that is used, as we have seen in the examples above contrasting CKA and angular Procrustes. Conversely if two models or datasets are compared then this comparison may result in different scores depending on the similarity measure. Prior work has used the 'condition average baseline' as a reference score for benchmarking models against neural data (Cloos et al., 2022). The condition average baseline is obtained by averaging each neuron's activity across trials from all experimental conditions, while still allowing each neuron to have its own unique time-varying firing rate, and then comparing this condition-averaged dataset to the original neural data. This baseline shows the similarity one can obtain by discarding the condition specific activity to keep only the condition independent dynamics of the neural activity. The condition average baseline is shown in Figures 2a and 2b as vertical dashed lines. For a single dataset, we are always comparing the same two quantities to obtain the condition average baseline but the scores are quite different depending on the similarity measure. For the angular Procrustes measure, as opposed to the linear regression similarity measure, this baseline score is closer to the point where task relevant information starts to be encoded.

## 2.2 Optimization dynamics of similarity scores

How much of the neural data must be captured by a synthetic dataset or model before the decode accuracy reaches the level seen in the neural dataset itself? One perspective on this question is to decode the task variables from neural data after projecting onto principal components 1 through N, where principal component 1 captures the most variance. In order to capture all the information about the task variables, at least several principal components must be included in the decode as shown in Figures 2c and 2d. This motivates the following hypothesis. Perhaps the reason that linear regression and CKA similarity scores can be so high while the synthetic data fails to encode task variables is because these similarity measures preferentially rely on the top few principal components. We explore this hypothesis in the following set of analyses with a synthetic dataset based on the neural recordings from Mante et al. 2013. Figure 3a shows the reference dataset (compare to Figure 1c). We can think of this reference dataset as a low-dimensional neural trajectory summarizing the population activity of many neurons, or alternatively, as the firing rates of two neurons over time (shown here encoding the two task variables of choice and dot motion coherence), recorded during six different experimental conditions, with the color in Figure 3a denoting the condition. Figure 3b shows the transformation of an initially random Gaussian noise dataset as it is optimized to maximize either the angular Procrustes or CKA similarity score with respect to the reference dataset. The score increases from an initial value near 0 to a maximum near 1 as optimization progresses, with the insets at the top of the figure showing the optimized noise dataset at various points during this procedure. The yellow curve shows how well the optimized dataset captures the first principal component of the reference dataset, as quantified by $R^2$, throughout optimization (see appendix A.2 for details). Notice that the second principal component, shown in purple, is only captured at a much higher optimization score for CKA versus angular Procrustes. Figure 3c shows the same re-

sults when a synthetic Gaussian noise dataset is optimized towards the reference dataset using either linear regression similarity or angular CKA similarity Williams et al. (2021).

The optimization dynamics not only depend on the similarity measure but also on the variance distribution of the reference dataset. Figure 3d shows four reference datasets with the same variance along the first principal component but decreasing variance along the second. If both dimensions have approximately equal variance then angular Procrustes, CKA, and linear regression will learn both dimensions similarly during optimization as shown by the white curve in Figure 3e. The curves are colored to indicate the fraction of variance the second principal component has relative to the first, so 1 indicates both principal components have the same variance. As the asymmetry between the principal components grows, the optimization to maximize angular Procrustes similarity still prioritizes the lower variance principal component (first column, second row), while CKA and linear regression do not capture the second principal component of the reference dataset until much later during optimization.

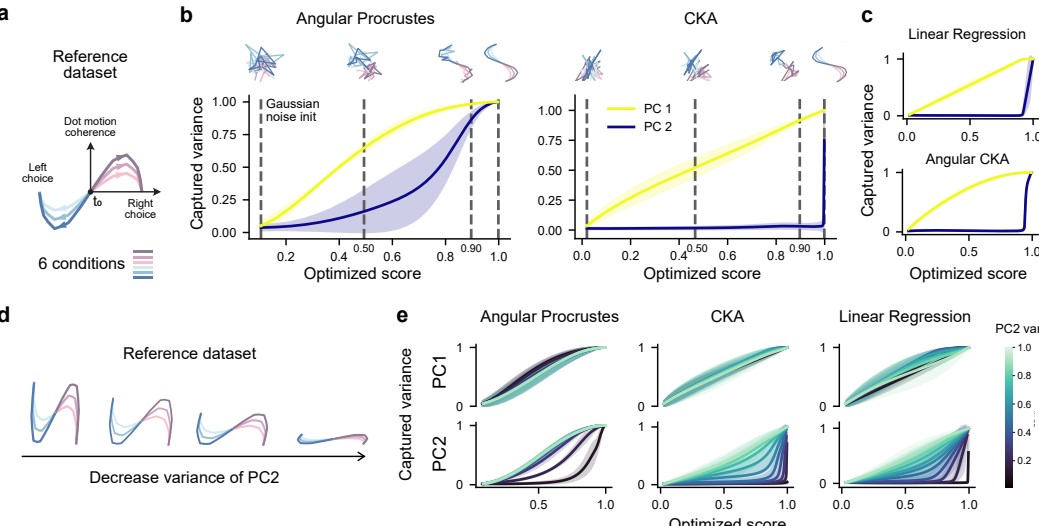

Figure 3: Different similarity measures differentially prioritize learning principal components of the data. (a) Reference dataset used as a target during optimization. (b, c) Initial Gaussian random noise data is updated to maximize similarity with the reference dataset, as quantified by one of the similarity measures. The transformation of the random noise dataset is shown at the top of panel b. The first principal component of the reference dataset is increasingly well captured by the optimized data as the similarity scores increase (yellow curves). The second, lower variance, component is also learned when maximizing the angular Procrustes similarity but is only captured at high similarity scores when maximizing linear regression, CKA, and angular CKA similarity. (d) Four reference datasets with decreasing variance along the second principal component. (e) Similarity measures capture both principal components when their variance is approximately equal. However, when the variance differs, CKA and linear regression have a much more pronounced preference for learning the high variance component as opposed to angular Procrustes which causes the optimized dataset to also capture the low variance dimension (curves colored according to asymmetry of variance distribution).

The optimization dynamics revealed in Figure 3a for a two-dimensional dataset also holds in higher dimensions. We now consider a 100-dimensional Gaussian reference dataset with an exponentially decaying eigenspectrum. Figure 4a shows the optimization dynamics when a 100-dimensional random noise Gaussian dataset is optimized towards this reference dataset by maximizing angular Procrustes similarity (top) or CKA similarity (bottom). Each curve shows how the optimized dataset captures a single principal component of the reference dataset during the course of optimization. Similar to the results in Figure 3, optimizing for angular Procrustes similarity captures more of the lower variance components in the data for a given similarity score. A convenient way of summarizing these curves is to note the similarity score required to capture a given principal component above some threshold, defined here (and shown in Figure 4a) as the centerpoint between the maximum and initial $R^2$ value for a given principal component. Figure 4b shows the score required

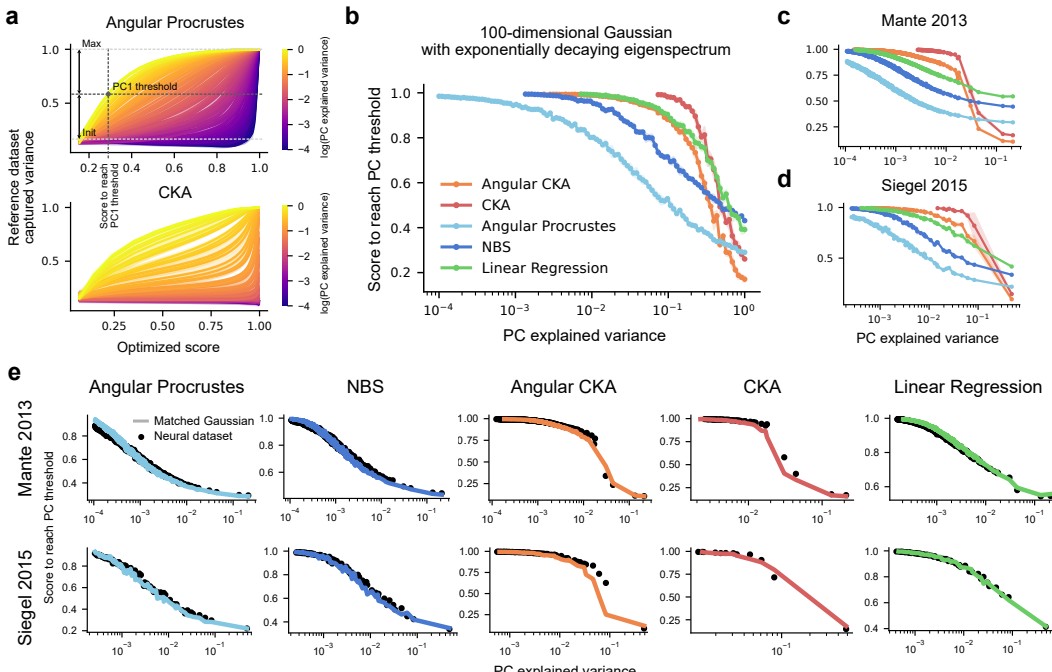

Figure 4: (a) A 100-dimensional standard Gaussian noise dataset is updated to maximize the similarity with a reference dataset, taken here to be a 100-dimensional Gaussian noise dataset with an exponentially decaying eigenspectrum. The principal components (PCs) of this reference dataset are captured by the optimized dataset at different similarity scores, which in subsequent figures we call the score to reach the PC threshold. (b) The score to reach the PC threshold is shown as a function of the variance explained by each PC. The highest variance PC is learned first during optimization at the lowest similarity score (bottom right of figure). A vertical slice through the figure shows the similarity score required to capture a specific PC. For example, to capture the PC at $10^{-1}$ requires a much lower similarity score when maximizing angular Procrustes versus linear regression (light blue curve is below the green curve). (c, d) The reference dataset used as a target during optimization is the neural activity from Mante et al. 2013 (panel c) and Siegel et al. 2015 (panel d). (e) The neural curves are the same as in panels c and d. The similarity scores at which PCs of this neural activity are learned, is well predicted by replacing neural activity with random Gaussian datasets that have a matching distribution of variances for each PC.

to reach the principal component threshold for the different principal components in the reference dataset. Figures 4c and 4d show the same curves when the reference datasets are now the electrode recordings from Mante et al. 2013 and Siegel et al. 2015. Surprisingly, the optimization dynamics shown in these figures is well matched when the reference neural dataset is replaced by a new reference dataset that consists of random Gaussian numbers with variances for each principal component matched to the neural data (Figure 4e). The variance distribution of the reference dataset strongly determines when each principal component is captured during optimization.

(Williams et al., 2021) proposed to take the arccosine of CKA to obtain a measure that satisfies the axioms of a distance metric. Figures 4b, 4c, and 4d additionally show that the metric version of CKA (referred to as angular CKA) improves its sensitivity to lower variance components. (Harvey et al., 2023) mathematically derived a similar relationship between angular Procrustes and Normalized Bures Similarity (NBS) (Tang et al., 2020), where the former can be obtained by taking the arccosine of the later. Again, we found that the arccosine increases sensitivity to lower variance components (Figures 4b, 4c, and 4d). (Harvey et al., 2023) also showed that NBS and CKA (and therefore angular Procrustes and angular CKA) essentially differ by a choice of matrix norm (see appendix A.1). NBS quantifies similarity using the nuclear norm, which involves a sum of singular values, whereas CKA uses the Frobenious norm, which sums the square of the singular values. The additional square operation in CKA significantly increases the contribution of large variance components, which is consistent with our results that CKA scores are mainly driven by these components.

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

# A APPENDIX

## A.1 SIMILARITY MEASURES

Before applying the similarity measures, the datasets are preprocessed by averaging the firing rates across trials with the same condition. The data of shape (time, conditions, neurons) is reshaped into a two-dimensional array by concatenating the time and condition dimensions together, resulting in an activity array of shape (time $*$ conditions, neurons).

Several methods have been proposed to measure similarity between models and neural data (Sucholutsky et al., 2023), (Klabunde et al., 2023). Although the approach presented here can be applied to any differentiable similarity measure, we selected three commonly used methods, linear regression (Yamins et al., 2014; Schrimpf et al., 2018), CKA (Kornblith et al., 2019), and Procrustes distance (Williams et al., 2021; Ding et al., 2021). We take the convention that an increasing score corresponds to increasing similarity, with a maximum value of 1 for perfect similarity.

Linear regression consists in finding the best linear mapping to predict a reference dataset $X$. We use $R^2$ as a measure of goodness of the linear prediction.

$$R_{LR}^2 = 1 - \frac{\min_B \|X - YB\|_F^2}{\|X\|_F^2}$$

Given that the columns of both $X$ and $Y$ are mean-centered, CKA can be formulated as:

$$\text{CKA}(X,Y) = \frac{\|X^T Y\|_F^2}{\|XX^T\|_F \|YY^T\|_F}$$

CKA varies between 0 and 1 (perfect similarity). (Williams et al., 2021) showed that the arccosine of CKA satisfies the axioms of a distance metric. We define a similarity score version of this angular CKA distance by first normalizing the distance between 0 and 1 and substracting the result from 1.

$$\text{AngularCKAScore}(X,Y) := 1 - \frac{\arccos(\text{CKA}(X,Y))}{\pi/2}$$

The angular CKA score varies between 0 and 1 (perfect similarity).

As shown in (Harvey et al., 2023), the Procrustes distance proposed in (Williams et al., 2021) is closely related to another measure, Normalized Bures Similarity (NBS) (Tang et al., 2020). In particular, Procrustes distance is the arccosine of NBS. Similarly to the score version of CKA distance defined previously, we define a score version of Procrustes distance.

$$\text{NBS}(X,Y) = \frac{\|X^T Y\|_*}{\sqrt{\|XX^T\|_* \|YY^T\|_*}}$$

where $\|\cdot\|_*$ denotes the nuclear norm of a matrix (also called the Schatten 1-norm), which is given by the sum of a matrix's singular values.

$$\text{AngularProcrustesScore}(X,Y) := 1 - \frac{\arccos(\text{NBS}(X,Y))}{\pi/2}$$

The angular Procrustes score varies between 0 and 1 (perfect similarity).

## A.2 OPTIMIZING SIMILARITY SCORES

Given a reference dataset $X$ with dimensions of (time, conditions, neurons), for example, we reshape it to a two-dimensional array by concatenating the time and condition dimensions. We initialize a synthetic dataset $Y$ with the same shape as $X$ and we sample each entry from a random standard Gaussian distribution. We use Adam (Kingma & Ba, 2017) to optimize $Y$ to maximize the similarity score with $X$, leveraging the differentiability of the similarity measures.

We use logistic regression to decode task variables as the synthetic dataset is optimized towards the reference dataset. We fit a different decoder for each time step and cross-validate using Stratified K-Folds. The reported decoding accuracy is the average test accuracy across time and across 5 splits.

We measure how well each PC, denoted by $v_i$, of the reference dataset $X$ is captured by the fitted data by (a) using linear regression to find the linear alignment that predicts the reference dataset best, and (b) measuring the captured variance by taking the $R^2$ of the error projected on each PC.

$$R^2_{PC_i} := 1 - \frac{\|v_i^T(X - \hat{X})\|^2}{\|v_i^T X\|^2}$$

where $\hat{X} = Y\hat{B}$ and $\hat{B} = \text{argmin}_B \|X - YB\|_F^2$.

