# OpenReview forum: "Differentiable Optimization of Similarity Scores Between Models and Brains"
_ICLR.cc/2024/Workshop/Re-Align — ICLR 2024 Workshop Re-Align Poster_

### Official Review · Reviewer_2Ltr · 2024-02-21
**Very nice and timely analysis**

**Rating:** 3
**Fit:** 3
**Confidence:** 3

**Workshop Review:**

There are now many dissimilarity measures that are proposed for comparing neural representations. Yet, there is very little guidance about how one should actually choose among the available options, and many people in the field seem to think these measures are all mostly equivalent. This paper shows that this common understanding is wrong, at least when comparing to biological datasets. The authors devise some very clever experiments to show that CKA and linear regression have a tendency to over-weight the top principal components of neural activity, relative to Procrustes distance. This difference could have very significant implications for comparing neural data to artificial network representations.

I would be interested to see how permutation Procrustes or the ["soft matching distance" described here](https://arxiv.org/abs/2311.09466) compares to the metrics investigated by the authors.

I would love to see this be highlighted as a contributed talk. As mentioned above, I have been largely dissatisfied with the lack of clear comparisons amongst metrics in the literature, and this paper is a breadth of fresh air to me.

**Reason For Not Giving Higher Score:**

N/A

**Reason For Not Giving Lower Score:**

This paper is asking the right questions and coming up with interesting results. The main weaknesses are that (a) it seems like the project is still being fully fleshed out since the paper is only 5 pages long, and (b) related to point a, there is a lot packed into 5 pages so the paper is dense and bit difficult to read without sub-sections and more organized writing. Nonetheless, I think the strengths outweigh these weaknesses.

**Reviewer Domain:**

neuroscience

---

### Official Review · Reviewer_3sCT · 2024-02-22

**Rating:** 2
**Fit:** 3
**Confidence:** 1

**Workshop Review:**

The paper compares commonly used similarity metrics (linear regression, CKA and  Procrustes distance) by making the similarity metric optimised dataset and analysing if they reflect task-relevant variables well.

Strong points:
The question addressed is important and it offers insights on similarity metrics used for representation comparison.
The results shown are interesting and experiments are well designed to demonstrate the arguments.

Weak points:
The manuscript can be more organised by subsectioning.
The results are interesting but some explicit notes on its implication might be useful.

I will accept the paper as it is well-aligned with the workshop scope and it targets important question in the field.
My major question lies around the optimization of the dataset with similarity metric - does this optimised dataset covers well all possible configurations of the dataset with high similarity metrics? If it is a trivial question to answer, I would appreciate explicit explanation in the manuscript and if not, it will be convincing to argue if this holds for other configurations as well.
I found the results interesting and I would like to know more about what is the practical implication.

The readability can be improved by subsectioning the later part of the paper.

**Reason For Not Giving Higher Score:**

N/A

**Reason For Not Giving Lower Score:**

N/A

**Reviewer Domain:**

neuroscience

---

### Official Review · Reviewer_av5m · 2024-02-24
**Similarity measures by optimizing synthetic data**

**Rating:** 1
**Fit:** 3
**Confidence:** 2

**Workshop Review:**

The authorss aim to evaluate similarity measures like linear regression, CKA, and Procrustes distance by optimizing synthetic datasets to resemble neural recordings. This work explores if these datasets can reflect neural activity without encoding task-relevant variables. The study finds discrepancies in how different measures capture task variables, highlighting the complexity of comparing model and brain representations.

**Reason For Not Giving Higher Score:**

1. The paper primarily focuses on a theoretical examination of similarity measures without providing clear implications for practical applications in neural modeling or computational neuroscience.
2. The paper's reliance on synthetic datasets optimized for similarity might not adequately represent the complexities of real neural data, limiting the generalizability of the findings.
3. The process of optimizing synthetic datasets and the criteria for evaluating similarity measures could be more transparent, making the methodology difficult to replicate or apply in different contexts.
4. The paper could benefit from a more thorough discussion of the limitations of its approach, particularly in terms of the representational capacity of synthetic datasets compared to actual neural data.

Citations to include:
Richards, B. A., et al. (2019). A deep learning framework for neuroscience. Nature Neuroscience, 22(11), 1761–1770. This reference can help situate the paper within the broader context of deep learning applications in neuroscience.

**Reason For Not Giving Lower Score:**

1. The paper introduces a unique perspective on evaluating similarity measures by optimizing synthetic datasets.
2. The analysis of different similarity measures and their capability to capture task-relevant variables is comprehensive..

**Reviewer Domain:**

machine learning

---

### Decision · Program_Chairs · 2024-03-02

Accept (Poster)